# Thermo-Optical Tuning Cascaded Double Ring Sensor with Large Measurement Range

**DOI:** 10.3390/s20185149

**Published:** 2020-09-09

**Authors:** Zhiping Yang, Yanlu Wang, Chang Su, Liyang Shao, Jian-Jun He, Mingyu Li

**Affiliations:** 1Department of Optical Engineering, School of Opto-Electronic Engineering, Changchun University of Science and Technology, Changchun 130022, China; zhipingyang7911@163.com (Z.Y.); yanluwang0@163.com (Y.W.); 2State Key Laboratory of Modern Optical Instrumentation, Centre for Integrated Optoelectronics, College of Optical Science and Engineering, Zhejiang University, Hangzhou 310027, China; suchang@ioe-zju.org (C.S.); jjhe@zju.edu.cn (J.-J.H.); 3Department of Electrical and Electronic Engineering, Southern University of Science and Technology, Shenzhen 518055, China; shaoly@sustc.edu.cn

**Keywords:** optical waveguide, cascaded double micro-ring resonator, thermo-optical tuning, intensity interrogation

## Abstract

In this paper, a thermo-optic tuning optical waveguide sensor system based on a cascaded double micro-ring resonator is investigated. The system consists of a micro-ring resonator with the microheater as a reference ring and a micro-ring resonator with removing the upper cladding layers as a sensing ring, combined with a microfluidic control. The refractive index change of the sample is measured by the electric power change of the microheater. The experimental results show that the sensitivity of the thermo-optic tuning is 34.231 W/RIU (refractive index units), and the measurement range is 4.325 × 10^−3^ RIU, almost eight times larger than that of the cascaded double micro-ring resonator without thermo-optic tuning for the intensity interrogation.

## 1. Introduction

Integrated optical sensor devices have been widely used in micro-ring resonator [1,2], photonic crystal (PhC) [3], and Mach-Zehnder interferometers (MZI) [4] due to their high integration and miniaturization. The bandgap of PhC sensors is formed by the periodic dielectric structure [3]. However, a high-precision photolithography process is required to make the feature pattern of the device, which is very unfavorable for commercialization. Based on the MZI structure sensor, highly sensitive power sensors and a stable system are required in the intensity interrogation to achieve the high sensitivity and the low level of noise [4]. Micro-ring resonators made of silicon on insulator (SOI) materials have been used widely in biosensing due to their high Q value and high sensitivity. SOI has a very high refractive index contrast that allows us to bend the waveguide very tightly with a radius of a few micrometers. Due to the high evanescent field on the surface of the silicon waveguide, the sensitivity of the SOI sensor is high [5,6]. The fabrication process of the SOI sensor is compatible with the complementary metal oxide semiconductor (CMOS) process. This means the SOI sensors can be made a large array for different functions, not only to detect the different type biomolecules [7,8,9,10] but, also, to use some sensors as a reference to monitor the temperature change [11,12].

Recently, a new type of electric tracking-assisted dual micro-ring optical sensor was reported [13]. The dual micro-ring sensor includes a sensing ring and a reference ring with an electrical controller. The shift of the resonance wavelength is converted into the change of the electric power by tuning the thermal heater. The radii of both micro-rings are designed with 10 μm, and the sensitivity of the sensing system is 15 mW/RIU (refractive index units), based on transverse electric (TE) polarized light. Such sensing systems can provide a low detection limit of 3.9 × 10^−6^ refractive index units (RIU) and have been used in biological detection. For the intensity interrogation of the cascaded double micro-ring resonators, a low-cost broadband light source and the optical power meters were used without measuring the spectrum [14,15,16]. However, the measurement range of the intensity interrogation is limited by the linear region of the output power with the refractive change of the analyte, and the detection limit is limited by the sensitivity of the power meter, which is an order of magnitude lower than the wavelength interrogation. Another intensity-sensing scheme uses a cascade micro-ring configuration based on the rib waveguide and tracks the maximum spectral position by using peak tracking methods [17,18]. This scheme was demonstrated to obtain an entire free spectral range of the probing micro-ring with the same radii of micro-rings. A significant shortcoming of this sensor scheme is that the limit of detection limit of this device is affected by the intensity peak width and provides a low detection limit (LOD) of 4.6 × 10^−5^ RIU. We address the detection limit by setting different radii micro-rings to produce a Vernier effect to improve. The waveguide was chosen as the slab waveguide for the transverse magnetic (TM) mode, and the sensor sensitivity is effectively improved in our scheme.

In order to achieve the low-cost, large measurement range and high sensitivity on-chip sensor, a thermo-optic tuning cascaded double ring (TTCDR) sensor is proposed in this paper, with a microheater on the reference ring for converting the output intensity changes into the electric power changes. The two micro-rings are designed to have different radii to produce a Vernier effect to increase the sensitivity. For a certain output intensity of the TTCDR sensor, the curve of the electric power changes with the different concentrations that the NaCl solution is plotted. The sensitivity of the sensor is 34.231 W/RIU for the TM mode. Utilizing the thermo-optic effect, the measurement range is almost eight times larger than that of the traditional cascaded double ring sensor without a microheater for the intensity interrogation.

## 2. Operating Device and Principle

Intensity interrogation and wavelength interrogation are two typical sensing methods for the cascaded micro-ring resonator sensors [14,15,16,17,18,19,20]. Intensity interrogation is spectrometer-free, which requires a low-cost broadband light source and a power meter. The refractive index change of the sample can be detected by the change of the output intensity of the sensor.

A schematic illustration of the TTCDR sensor is shown in Figure 1. As shown in Figure 1, the sensor is comprised of two main components: a sensing ring without an upper-cladding layer and a reference ring with a microheater. There is a titanium nitride (TiN) microheater on the reference ring for thermo-optic tuning. We adopt a low-cost broadband light source as the light source (Conquer, OS-ASE-M2-C-0-100-0-S-FA). This source was provided by Beijing Conquer Optical Technology Co. Ltd., Beijing, China. The light source has the advantages of a high output power, spectral line width, low degree of polarization, high power stability, and good average stability, which can meet the stringent performance requirements of the broadband light source (BLS) for sensing and testing.

The BLS is connected to the attenuator and tunable pass band filter (TPBF) in Figure 1. Then, the light is coupled into the input and output of the sensor chip by the grating couplers. The input (port 1) and output (port 2 and port 3) shared the same fiber array for easy reuse. Finally, the output power was collected by the power meter (Agilent 81634A). The current source (KEITHLEY 2400) is controlled by PC for the microheater and connected to the sensor through a probe. The output through and drop of the sensor were received by detector 1 and detector 2, respectively, and finally, the computer received the data. The windowed sensing ring was in direct contact with the substance to be measured through the microfluidic channel to sense the change in the refractive index and realize sensing.

The peak wavelength shift in the envelope function of the cascaded double ring resonators transmission is magnified by a Vernier effect factor. The free spectrum range (*FSR*) of the output transmission *FSR_output_* is given as [15]:(1)$$FSR_{output}=\frac{FSR_{sen}\times FSR_{ref}}{\left|FSR_{sen}-FSR_{ref}\right|}$$
where *F* is the Vernier effect amplification factor, *F*
$=\frac{FSR_{ref}}{\left|FSR_{sen}-FSR_{ref}\right|}$, and *FSR_sen_* and $FSR_{ref}$ are the *FSR* of the reference ring and sensing ring, respectively.

The operating principle of the TTCDR sensing system is shown in Figure 2. When the refractive index of the analyte changes Δ*n,* the output intensity of the TTCDR sensor changes Δ*I*. The sensitivity of the cascade double micro-ring sensor (CMRR) is defined as *S_CMRR_* = Δ*I/*Δ*n*. In Figure 1, in order to obtain the high sensitivity, the bandwidth of the TPBF is equal to half of *FSR* the of the cascade double rings transmission envelope. The spectrum of the BLS is *I_In_* (*λ*), and *λ* is the wavelength. The output intensity *I_out_* of the TTCDR sensor can be expressed as [15]:(2)$$I_{out}=\int_{0}^{\infty }[I_{In}\left(\lambda \right)T_{ref}\times T_{sen}]d\lambda$$
where $T_{ref}$ and $T_{sen}$ are the transmission spectra of the reference ring and the sensing ring port, respectively.

The thermo-optic tuning principle is shown in Figure 2. When the refractive index changes by Δ*n*, the transmission spectrum shifts from a black line to a blue line, as shown in Figure 2c. The output spectrum (Figure 2d) is the product of the blue line transmission spectrum of Figure 2c and the broadband light source spectrum (red line). The output spectrum changes from the Figure 2b to the Figure 2d when the refractive index of the analyte changes Δ*n*. When the microheater is applied by an electric power Δ*P_elec_*, the transmission spectrum of the Δ*n* refractive index change shifts from a blue line to a black line in Figure 2e, and the envelope of the transmission can return to the initial one of Figure 2b, as shown in Figure 2f. The sensitivity of the TTCDR sensor can be expressed as *S* = Δ*P_elec_*/Δ*n*.

When the refractive index of the analyte changes, the electric power of the microheater is scanned, until the Δ*I_out_*/Δ*P_elec_* is equal to the initial value and Δ*I_out_* = 0. Therefore, by measuring, the power change of the microheater Δ*P_elec_* can detect the variation of the sample refractive index Δ*n*.

## 3. Experiments and Results

### 3.1. Fabrication of TTCDR Chip and Characterization

The wafer preparation and technical assistant was by the Integrated Circuit Advanced Process Center (ICAC) of the Institute of Microelectronics of Chinese Academy of Sciences (IMECAS), Beijing, China. The fabrication started by spin-coating a thin film of photoresist on the SOI wafer. Waveguides, grating couplers, and micro-rings were fabricated by the stepper, then growing 2-μm SiO_2_ upper cladding, and the sensing window was opened by photolithography. Finally, the resistance heating metal TiN and the conductive electrode Al layer are coated by the sputtering. The micro-ring chip was fabricated on a silicon insulator platform with a 220-nm-thick silicon top layer and a 2-μm-thick buried oxide layer. The SOI micro-ring sensor is based on the evanescent field. The TE mode used for sensing has one order lower sensitivity compared to the TM mode, due to the smaller mode-field overlap with the sample solution [21]. The silicon strip waveguide is designed with the width of 550 nm for the TM mode.

The whole chip was covered by the SiO_2_ upper cladding layer, except that the sensing ring is exposed to the reagent sample by removing the upper cladding layer in the sensing window. The optical microscope image of the sensor chip is shown in Figure 3a, and the size of the whole chip is 2 cm × 2 cm. The scanning electron microscopy (SEM) images of the sensing ring and the directional coupler between the bus waveguide and the ring are shown in Figure 3b,c, respectively. The image of the TTCDR sensing system consists of the microfluidic channels, the sensor chip, the microheater probes, and the fiber arrays, as shown in Figure 4a. In Figure 4b is a SEM image of the TiN microheater.

### 3.2. Intensity Interrogation with Thermo-Optic Tuning

To produce the Vernier effect, the radii of the reference ring and the sensing ring are 123 μm and 121 μm, respectively. The optical transmission spectra of the drop port was measured in Figure 5 with the refractive index of $1.8\times 10^{-3}$ RIU (1.0% NaCl solution, in black) and $2.7\times 10^{-3}$ RIU (1.5% NaCl solution, in blue). The refractive index of the NaCl solution varies by $1.8\times 10^{-3}$ per 1% concentration of variation [22]. The concentration of NaCl with their refractive index is listed in Table 1. The red curve in Figure 5 shows the spectrum of the input BLS through the TPBF. The *FSR_output_* of the transmission curve envelope is 11.92 nm. To achieve the highest sensitivity, the input BLS with a 3-dB bandwidth of ~6 nm and a center wavelength of 1543.81 nm was chosen by the TPBF. The peak of the transmission spectral envelope has a blue shift with increasing the concentration of NaCl for their refractive index from $1.8\times 10^{-3}$ RIU (1.0%) to $2.7\times 10^{-3}$ RIU (1.5%).

In order to eliminate the influence of the BLS fluctuations, the output intensity of drop port 3 is normalized by the output power through port 2. The normalized output power changes with the electric power of the microheater periodically, as shown in Figure 6a, under a refractive index of $1.8\times 10^{-3}$ RIU (1.0% NaCl solution). When the concentration of the NaCl solution with their refractive index changes from $1.8\times 10^{-3}$ RIU (1.0%) to $7.2\times 10^{-3}$ RIU (4.0%), the responses of the sensor are measured four times at each the refractive index changes of the concentration of NaCl solutions, as shown in Figure 6b without the electric power of the microheater. The normalized output power decreases with increasing the concentration of the NaCl solution for their refractive index. The output intensity changes periodically with the different concentration of NaCl solution for their refractive index. In Figure 6b, the experimental results showed that the measurement range for the TTCDR sensor Range 2 = 4.325 × 10^−3^ RIU was almost eight times larger than that of the traditional cascaded double ring sensor Range 1 = 0.54 × 10^−3^ RIU for the intensity interrogation. Range 1 is determined by Δ*I*/Δ*n*, and Range 2 is determined by Δ*I*/Δ*n* and Δ*I* = 0. In the intensity interrogation, the sensing is realized by the slope of intensity variation (Δ*I*/Δ*n*) [14], and the intensity detection range needs to be in the linear region of the power curve (Figure 6b) to obtain high sensitivity in the sensing experiment. Thermal tuning can solve the problem of the nonlinear area of the power curve, the detection range can cover the total range of power curve, and every point on the curve is linear.

The choice of the initial point is very important in the experiment. We set the starting point at the maximum slope of the power curve instead of the peak intensity of reference [17,18]. Since the slope of the power curve is equal to zero at the peak intensity, it is very difficult to find the peak position accurately. This makes the uncertainty of the peak point relatively large and affects the detection limit of the sensor. In Figure 6b, the Δ*I*/Δ*n* of the initial point m is not equal to 0. The shift of the transmission curve due to the variation of the different concentration solutions can be converted to the electric power change of the heater by keeping the slope of the output power curve and the output power as the constant, as shown in Figure 7.

The sensitivity of the thermo-optical tuning cascade micro-ring is 34.231 W/RIU, as shown in Figure 7. Accordingly, the limit of detection (LOD) for the measurement of the refractive index of the sensor is calculated based on the standard deviation *σ* = 0.105 mW divided by the sensitivity S: LOD = 3*σ*/S = 9.202 × 10^−6^ RIU. The standard deviation *σ* is obtained from the response measurement from 1.5% to 1.0%. The intensity interrogation method of the traditional cascaded double ring sensor can be improved by using thermo-optic tuning. No matter where the initial operating point is located, the electric power of the microheater is linear with the refractive index change of the NaCl solution.

The micro-ring resonator sensor based on the SOI waveguide has high refractive index contrast. Most of the electric field is distributed in the cladding of the waveguide. Due to the small overlap with the mode field of the analyte, the TE mode used for sensing has a low sensitivity. The TM mode electric field is distributed on the surface of the waveguide, and the strong surface electric field can make the sensing substance have a greater influence on the effective refractive index of the waveguide, and it is easier to achieve a high sensitivity [21]. Sensors are made of rib waveguide materials [18], the electric field is mainly distributed inside the core layer, and the sensitivity of the sensor is relatively low: S = 3*σ_p_*/LOD = 3.021 W/RIU (*σ_p_* = 4.638 × 10^−5^ mW and LOD = 4.606 × 10^−5^ RIU). In our TTCDR sensing system, the cascaded ring resonators are constructed by the slab waveguides. The two micro-rings are designed to have different radii to produce a Vernier effect so as to increase the sensitivity. The experiments showed a high sensitivity of S = 34.231 W/RIU for the TM mode. The LOD was also improved: LOD = 9.202 × 10^−6^ RIU. Keeping Δ*I_out_*/Δ*P_elec_* and *I_out_* as the constants are the determination condition of the electric power change, instead of using the intensity peak tracking method in order to reduce the uncertainty of the sensor. In addition, the detection limit and sensitivity of the micro-ring resonator can also be improved by using a high-resolution current source.

## 4. Conclusions

An optical TTCDR sensing system based on the SOI substrate has been fabricated by the CMOS process. The TTCDR sensing system includes the BLS, attenuator, TPBF, power meters and sensing chip integrated with a grating coupler, a reference ring, a microheater, and a sensing ring. The analyte can be measured by the electric power change of the microheater without requiring the spectral measurement to reduce the cost. The experimental results show that the sensitivity is 34.231 W/RIU, the detection limit is 9.202 × 10^−6^ RIU, and the measurement range is 4.325 × 10^−3^ RIU. The TTCDR sensing system has a great potential for biosensing applications.

## Figures and Tables

**Figure 1 sensors-20-05149-f001:**
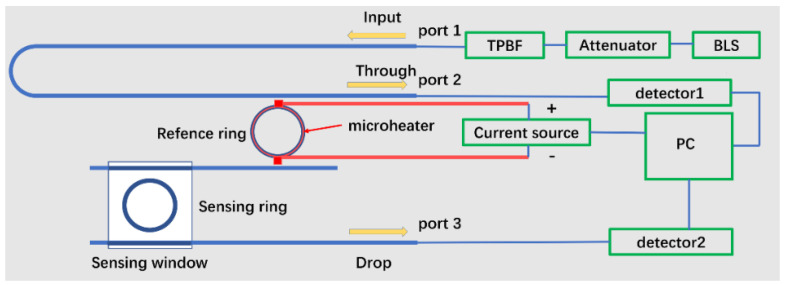
Schematic image of the total sensing system. BLS: broadband light source. TPBF: tunable pass band filter. PC: personal computer.

**Figure 2 sensors-20-05149-f002:**
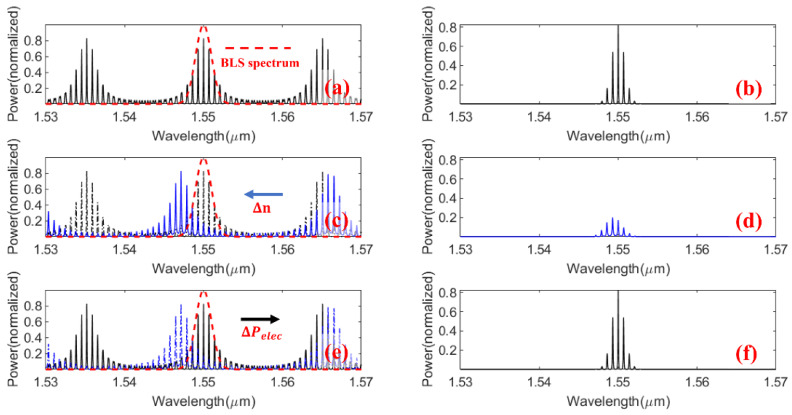
Operating principle of the thermo-optic tuning cascaded double ring (TTCDR) sensor. (**a**) is the transmission spectrum and the spectrum of the broadband light source (dashed red line) without the refractive index changes, and (**b**) the output spectrum is the product of the BLS spectrum and the transmission spectrum. (**c**) When the refractive index changes Δn, the transmission spectrum shifts from dashed black line to the solid blue line. (**d**) The output spectrum. (**e**) When the microheater is applied by an electric power Δ*P_elec_*, the transmission spectrum shifts from the dashed blue line to the solid black line. (**f**) is the output spectrum for the electric power Δ*P_elec_* change.

**Figure 3 sensors-20-05149-f003:**
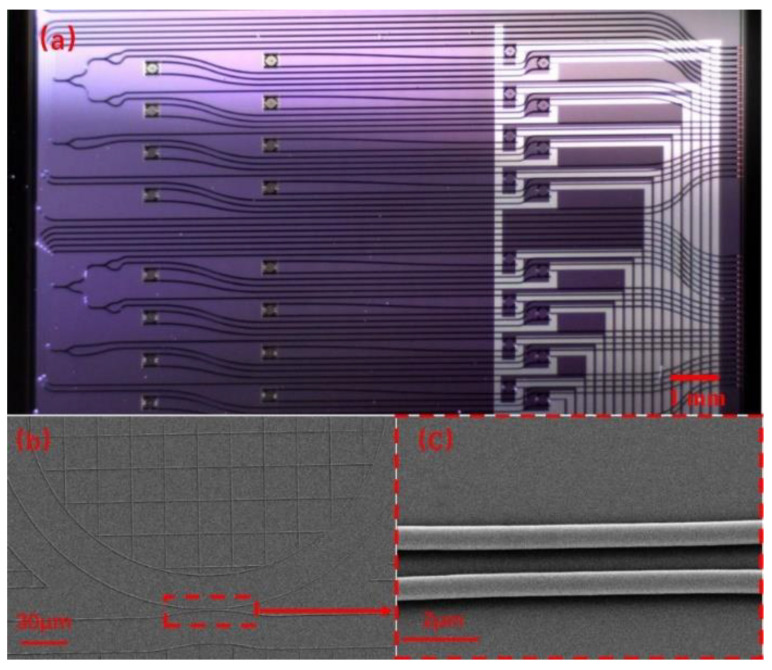
(**a**) Optical microscope image of the sensor chip. (**b**) SEM image of the sensing ring. (**c**) SEM image of the directional coupler.

**Figure 4 sensors-20-05149-f004:**
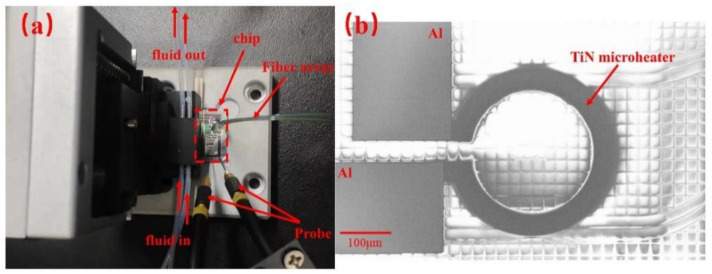
(**a**) Figure of the microflow control system. (**b**) SEM image of the TiN microheater.

**Figure 5 sensors-20-05149-f005:**
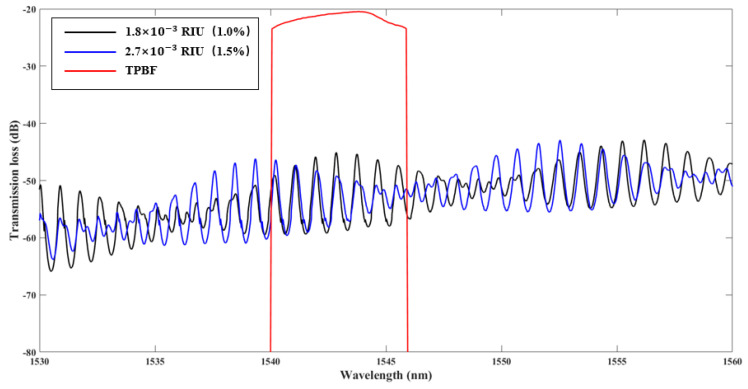
Experimental transmission spectrum of the drop port with the refractive index of $1.8\times 10^{-3}$ refractive index units (RIU) (1.0% NaCl solution), 2.7 × 10^−3^ RIU (1.5% NaCl solution), and the spectrum of the input BLS through TPBF.

**Figure 6 sensors-20-05149-f006:**
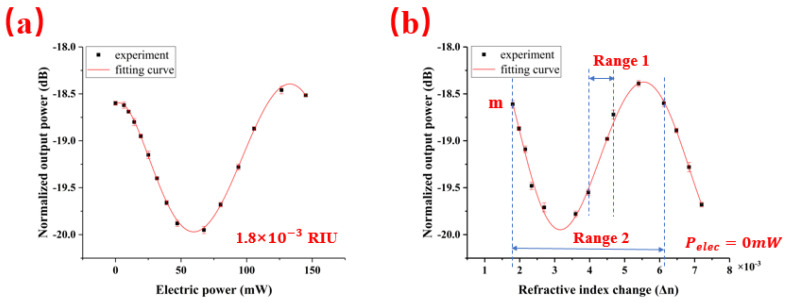
(**a**) The fitting curve of the normalized output power varies with the electric power under the refractive index of RIU (1.0% NaCl solution). (**b**) Measured normalized output power versus the refractive index change of NaCl solutions with different concentrations and a fitting curve based on 0-mW electric power.

**Figure 7 sensors-20-05149-f007:**
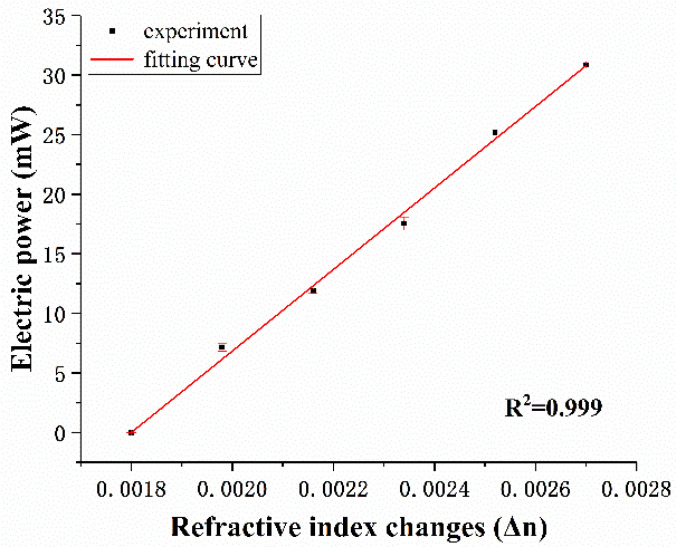
The fitting curve of the electric power changes versus the refractive index changes of the different NaCl solution concentrations.

**Table 1 sensors-20-05149-t001:** The concentration of NaCl with their refractive index [22].

Concentration of NaCl Solution	Refractive Index Units
1.0%	1.8 $\times 10^{-3}$ RIU
1.1%	1.98 $\times 10^{-3}$ RIU
1.2%	2.16 $\times 10^{-3}$ RIU
1.3%	2.34 $\times 10^{-3}$ RIU
1.4%	2.52 $\times 10^{-3}$ RIU
1.5%	2.7 $\times 10^{-3}$ RIU
2.0%	3.6 $\times 10^{-3}$ RIU
3.0%	5.4 $\times 10^{-3}$ RIU
4.0%	7.2 $\times 10^{-3}$ RIU

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
