# Peer review of "Thermo-Optical Tuning Cascaded Double Ring Sensor with Large Measurement Range"

_sensors, 2020, doi:10.3390/s20185149_

Round 1
Reviewer 1 Report
Authors present a therma-optical tuning cascade double ring resonator sensor. One of the ring works as reference and has integrated a microheater to regulate the refractive index of the sample. The other ring resonator has an open window on the top cladding for sensing applications. Authors present the design, fabrication and characterization of the sensor.
They did a good introduction, where I miss some references and some comments about other integrated optics design, as interferometric or photonic crystal sensors. Also, in line 37 in page 1 I recommend change 3.9m-RIU for 3.9 · 10-6 RIU, I think is clearer for the reader.
I recommend clarify Figure 2. Authors should describe what represent blue line in the graph. It is not clear to me the difference between Figure 2-c and Figure 2-e. I understand the first one is when a refractive index change is applied and the other one when the microheater is applied, but I don't understand what represent blue and black line on these graphs. Also, why there are black line in Figure 2-b and Figure 2-f, and there is blue line in Figure 2-d.
In line 100 in page 3 in SIO2 should be change the number 2 by subscript
In section 3.2 I recommend to change the concentration of NaCl for their refractive index. Authors are investigating the refractive index change, not the concentration of NaCl solution on the paper.
I suggest checking the significant numbers throughout the article. In my opinion authors use many decimals. For example "9.202×10-6" or "4.325×10-3 RIU" in conclusions should be checked comparing with the accuracy of the measurements to know if 0,1 or 2 decimals should be enough.
Reviewer 2 Report
In this paper a thermo-optic tuning optical waveguide sensor system based on cascaded double micro-ring resonator was developed and characterized. A micro-ring resonator with microheater as a reference ring is used. The refractive index change of the sample is measured by the electric power change of the microheater. However, similar papers published before developed theoretically and experimentally (Thermo-optical tuning of cascaded double micro-ring resonators for dynamic range enhancement,” in Proc. SPIE, 2016, vol. 9725, no. 24, Full-Range detection in cascaded microring sensors using 207 thermooptical tuning. Journal of Lightwave Technology 2016,34, 5157-5163) the same sensor and obtained very similar results. It is true that in the present paper authors presented some improvement (6 times) regarding the measurement range, however detection limits of detection are very similar to the mentioned paper and even to others conventional based ring resonator sensors (10-5 RIU) (Label-free biosensing with a slot-waveguide-based ring resonator in silicon on insulator,” IEEE Photonics J., vol. 1, no. 3, pp. 197–204, 2009.) I did not see any impressive improvement regarding in this paper when compared to the state of the art publications. Moreover, no any discussion was presented nor in the introduction part neither in the discussion of results as a matter of comparison. This paper needs major corrections in order to clarify the ideas, based on my arguments above and detailed explained below:
- The title should be corrected. The correct written is thermo-optical instead thermal-optical.
- Abstract:
“the measurement range is 4.325×10-3 RIU, almost eight times larger than that of the cascaded double micro-ring resonator without thermo-optic tuning for the intensity interrogation.”
The results must be compared to the above mentioned papers in which the difference is not eight times.
- The introduction is poor and lacks important information. One of the most important papers were not discussed, however this were the first developments about detection in cascaded microring sensors using thermo-optical tuning (Thermo-optical tuning of cascaded double micro-ring resonators for dynamic range enhancement,” in SPIE, 2016, vol. 9725, no. 24, Full-Range detection in cascaded microring sensors using 207 thermooptical tuning. Journal of Lightwave Technology 2016,34, 5157-5163).
- The operating device and principle are poorly described, the elements presented in Fig. 1 should be detailed described.
- The fabrication of the micro-ship presented in figure 3 was described superficial, important information is missing about the techniques, devices and procedure for such fabrication.
- Figure 7 needs a detailed discussion and comparison with Fig. 2 of the above mentioned paper (Journal of Lightwave Technology 2016,34, 5157-5163).
- In general discussion of results must be provided deeper in order to understand the novelty of this paper when compared to the others in which were published the same device with almost same results (Thermo-optical tuning of cascaded double micro-ring resonators for dynamic range enhancement,” in SPIE, 2016, vol. 9725, no. 24, Full-Range detection in cascaded microring sensors using 207 thermooptical tuning. Journal of Lightwave Technology 2016,34, 5157-5163), if not this paper cannot be accepted.
Reviewer 3 Report
The authors report an experimental thermo-optic tuning optical waveguide sensor system based on a cascaded double microring resonator and discuss its advantages over similar microring-based sensors. I believe that results are interesting and reliable. But I have some comments and suggestion for the authors in order to improve the paper.
- BLS should be described in more detail. Is this a commercially available or a home-made device? Its spectrum should be plotted. The stability of BLS should be commented.
- How accurate is the obtained data (sensitivity of 34.231 W/RIU, the detection limit of 9.202×10-6 RIU, and the measurement range of 4.325×10-3 RIU)? Was the error estimated for them? This point should be discussed in the manuscript.
- The SEM model should be provided.
- Several abbreviations (RIU, CMOS, SEM) are used without explanation but should be given.
- Figure 6(b), x-axis: “Concentration of Nacl solution (%)”, should be replaced by “Concentration of NaCl solution (%)”. The same (Nacl-> NaCl) should be done in the figure caption.
Round 2
Reviewer 1 Report
Authors has followed most of the reccomendations of the first review, but they didn't change the follotowing comments:
- "In section 3.2 I recommend to change the concentration of NaCl for their refractive index. Authors are investigating the refractive index change, not the concentration of NaCl solution on the paper.". Maybe I am not sufficient clear with this comment. I recommend to change the value of the concentration of NaCl for it value in refractive index. For example, 1.0 % could be 1.421 and 1.5 % 1.432 (I am inventing these refractive index values, but authors should know these real value), to know the refracive index change. I would change these values in the whole section.
- I am not agree with the response over the last comment about the significant numbers. The final numbers of decimals not depends on the accurate of the power supply or the power meter. To know the final number of decimals you should do the propagation errors with all the variables and I am sure the final result won't show 3 decimals.
Reviewer 2 Report
The paper was improved, author answered the question properly emphasizing the novelty
